# Intranasally Inoculated SARS-CoV-2 Spike Protein Combined with Mucoadhesive Polymer Induces Broad and Long-Lasting Immunity

**DOI:** 10.3390/vaccines12070794

**Published:** 2024-07-18

**Authors:** Tomoko Honda, Sakiko Toyama, Yusuke Matsumoto, Takahiro Sanada, Fumihiko Yasui, Aya Koseki, Risa Kono, Naoki Yamamoto, Takashi Kamishita, Natsumi Kodake, Takashi Miyazaki, Michinori Kohara

**Affiliations:** 1Department of Microbiology and Cell Biology, Tokyo Metropolitan Institute of Medical Science, Tokyo 156-8506, Japan; honda-tm@igakuken.or.jp (T.H.);; 2Graduate School of Medical and Dental Sciences, Niigata University, Niigata 951-8510, Japan; 3Toko Yakuhin Kogyo Co., Ltd., Toyama 930-0211, Japan

**Keywords:** SARS-CoV-2 surface antigen, mucoadhesive polymer, carboxy-vinyl polymer (CVP), enhancement of mucosal immune responses, intranasal inoculation

## Abstract

Current mRNA vaccines against SARS-CoV-2 effectively induce systemic and cell-mediated immunity and prevent severe disease. However, they do not induce mucosal immunity that targets the primary route of respiratory infection, and their protective effects wane after a few months. Intranasal vaccines have some advantages, including their non-invasiveness and the additional ability to activate mucosal immunity. In this study, we aimed to explore the effectiveness of an intranasally inoculated spike protein of SARS-CoV-2 mixed with a carboxy-vinyl polymer (S–CVP), a viscous agent. Intranasally inoculated S–CVP strongly induced antigen-specific IgG, including neutralizing antibodies, in the mucosal epithelium and serum and cellular immunity compared to the spike protein mixed with aluminum potassium sulfate. Furthermore, IgA production was detected only with S–CVP vaccination. S–CVP-inoculation in mice significantly suppressed the viral load and inflammation in the lung and protected mice against SARS-CoV-2 challenges, including an early circulating strain and the Omicron BA.1 variant in a manner dependent on CD8^+^ cells and monocytes/neutrophils. Surprisingly, high antibody responses and protective effects against multiple variants of SARS-CoV-2, including Omicron BA.5, persisted for at least 15 months after the S–CVP immunization. Hence, we propose intranasal inoculation with S–CVP as a promising vaccine strategy against SARS-CoV-2.

## 1. Introduction

Coronavirus disease 2019 (COVID-19), caused by severe acute respiratory syndrome coronavirus 2 (SARS-CoV-2), presents a global public health crisis [1]. Various vaccines have been developed to mitigate the impact of SARS-CoV-2 infections [2]. Globally, mRNA vaccines, which induce a robust systemic immune response and exhibit prophylactic efficacy in terms of severe disease development, are used [3,4]. However, the SARS-CoV-2 virus mutates rapidly and several mutant strains have emerged that escaped vaccine-induced immunity, resulting in frequent breakthrough infections [5]. Neutralizing antibodies are highly effective in inhibiting viral growth; however, their efficacy declines within a few months of vaccination, thus, multiple boosters are required to maintain high levels of immunity [6]. Therefore, future efforts to develop vaccines should focus on inducing strong immunity with long-lasting effectiveness against a wide range of mutant strains. However, intramuscular administration of the vaccine does not effectively induce a respiratory mucosal immune response, which can effectively prevent respiratory infections, such as SARS-CoV-2 [7]. Moreover, the administration of these vaccines is often associated with invasive methods.

The efficacy of intranasal vaccination in inducing mucosal and systemic immunity has been well established. This strategy, associated with ease of administration and non-invasiveness, is considered advantageous over needle injection-based vaccination. Thus, intranasal inoculation is a potentially beneficial approach for administering vaccines against infectious diseases. However, intranasal vaccines require strong and safe adjuvants, which is attributable to the fact that many antigens are rapidly removed via mucociliary clearance. Additionally, retrograde transport to the brain occurs via the olfactory nerve [8]. Furthermore, various adjuvants, including toxoids and pattern-recognition receptor agonists, have been explored [9,10]. Cynomolgus macaques intranasally inoculated with inactivated influenza virus and carboxyvinyl polymer (CVP), exhibited an increased IgA response [11]. Moreover, intranasal inoculation with hepatitis B virus protein mixed with CVP induces neutralizing antibody production and interferon (IFN)-γ response [12].

CVP is an acrylic acid-based hydrophilic polymer with several applications; for instance, it is used as a thickening agent in cosmetics and pharmaceuticals. Hence, the safety profile of CVP is well-characterized. In this study, we characterized the humoral and cellular immune responses and the systemic immunity induced by the spike (S) protein of SARS-CoV-2 combined with CVP (S–CVP), intranasally inoculated in mice. Additionally, the protective efficacy of S–CVP was evaluated, as well as long-term immunity and cross-reactivity.

## 2. Materials and Methods

### 2.1. Ethics Statement

This study was conducted in accordance with the Guidelines for Animal Experimentation of the Japanese Association for Laboratory Animal Science and the recommendations of the Guide for the Care and Use of Laboratory Animals of the National Institutes of Health. All animal care and experimental procedures were performed according to the guidelines established by the Tokyo Metropolitan Institute of Medical Science Subcommittee on Laboratory Animal Care. All experimental protocols were approved by the Committee on Ethics of Animal Experiments of the Tokyo Metropolitan Institute of Medical Science (20-085, 21-080, 22-078, 23-079).

### 2.2. Animals and Vaccination

Female BALB/c mice (6-week-old) were purchased from Japan SLC, Inc. (Shizuoka, Japan). Purified His-tagged active trimer of SARS-CoV-2 S protein (the early pandemic strain; accession # QHD43416.1, Acro, #SPN-C52H8), diluted in PBS (−), was mixed with an equal volume (5 µL per animal) of CVP (Tokoyakuhin) and used to vaccinate mice. Mice were intranasally inoculated with the protein (5 µg per animal) combined with CVP; a dose of 2.5 µg of protein was injected into each nostril. S antigen mixed with alum was used as a control; for this purpose, 5 µg antigen mixed with 50 µg of alum was administered to each animal subcutaneously.

### 2.3. ELISA

The wells of 96-well plates were coated by incubating with 1 µg/mL of recombinant C-terminus His-tagged SARS-CoV-2 S protein (the early pandemic strain; Acro, #SPN-C52H8, B.1.1.529/Omicron; Acro, #SPN-C52Hz, or BA.4/BA.5/BA.5.2; Sino Biological, Beijing, China #40589-V08H32) dissolved in 50 mM carbonate buffer (pH 9.6) overnight at 4 °C, and thereafter treated with blocking buffer (PBS containing 1% bovine serum albumin, 2.5 mM ethylene diamine tetra-acetic acid, and 0.5% Tween-20) for 2 h at room temperature (RT). Diluted samples were added to the wells, and incubated overnight at 4 °C. After incubation, wells were washed with PBS (−)-T (PBS (−) containing 0.05% Tween 20), and horseradish peroxidase (HRP)-labeled anti-mouse IgG (1:8,000, Thermo Fisher #62-6520) or HRP-labeled anti-mouse IgA (1:10,000, Bethyl #A90-103P) was added to individual wells, and incubated for 2 h at RT. After washing, TMB solution (Beacle #BCL-TMB-21) was added to the wells, and incubated for 15 min. The reactions were quenched by adding 2 M sulfuric acid, and the absorbance at 450 nm was measured using an EnVision plate reader (Perkin Elmer, Waltham, MA, USA).

### 2.4. Viruses

SARS-CoV-2 early pandemic strains: JP/TY-WK-521/2020 (WK-521; Accession: LC522975), SARS-COV-2 Omicron BA.1 variant: hCoV-19/Japan/TY38-873/2021 (TY38-873; GISAID strain name: EPI_ISL_7418017), and SARS-CoV-2 Omicron BA.5 variant: hCoV-19/Japan/TY41-702, kindly provided by Drs. Masayuki Saijo, Mutsuyo Takayama-Ito, Masaaki Sato, and Ken Maeda, National Institute of Infectious Disease, were used as challenge strains. The viruses were cultured in VeroE6/TMPRSS2 cells (JCRB1819) and grown in DMEM (Nissui, Tokyo, Japan) supplemented with 10% inactivated fetal bovine serum and G-418 (1 mg/mL).

### 2.5. PRNT

Serially diluted mouse serum was incubated with 100 PFU of SARS-CoV-2 early pandemic strain (TY/WK-521), Omicron BA.1 (TY38-873) or Omicron BA.5 (TY41-702) variant for 1 h at 37 °C. Subsequently, TMPRSS2-expressing Vero cells were incubated with these mixtures on a 6-well plate for 1 h with rocking. The samples were then removed from the plate, and growth medium containing 0.6% agarose was added to cover the wells. After 2–3 days of incubation, the plate was fixed with formalin and the plaques were visualized with crystal violet. The highest dilution of the serum that reduced the number of viral plaques by 50% (PRNT_50_) was defined as the neutralizing antibody titer.

### 2.6. ELISpot Assay

Six-week-old female BALB/c mice were inoculated intranasally twice 4-weeks apart. Five weeks after the first inoculation, the mice were euthanized, and splenocytes were isolated. ELISpot assay was performed as described previously [13]. In brief, isolated splenocytes were seeded into the wells of 96-well plates coated with anti-IFN-γ (MABTECH, Nacka strand, Sweden #3321-2H), -IL-2 (MABTECH #3441-2H), or -IL-4 (MABTECH #3311-2H) antibody and subsequently stimulated with pooled SARS-CoV-2 S peptide (Pep Tivator SARS-CoV-2 Prot_S complete [130-127-951, Miltenyi Biotec] or Pep Tivator SARS-CoV-2 Prot_S B.1.1.529 WT [130-129-927, Miltenyi Biotec, Bergisch Gladbach, Germany]) for 48 h at 37 °C. The production of IFN-γ, IL-2, and IL-4 was analyzed using an ELISpot assay kit (MABTECH) according to the manufacturer’s instructions.

### 2.7. Infection

The recombinant SARS-CoV-2 S protein (5 µg per animal) mixed with or without CVP (single-dose) was intranasally administered to female BALB/c mice. As a control, a single or double dose of the mixture of S protein and alum was administered subcutaneously in a 4-week interval. Six weeks after vaccination, mice were infected with 1 × 10^5^ PFU/50 µL/animal of SARS-CoV-2 early circulating strain (TY/WK-521) or Omicron BA.1 variant (TY38-873); five days before the viral infection, mice were inoculated intranasally with 5 × 10^7^ focus-forming units (FFU) of adenoviruses expressing hACE2 [14] per animal. Body weight was monitored on a daily for 5 days and euthanasia was determined at a 30% loss of initial body weight. Animals were euthanized 5 days after infection, and blood and internal organs were collected for further analysis. The following schedule was used for long-term immunization: female BALB/c mice were inoculated intranasally with purified SARS-CoV-2 S protein (5 µg per animal) combined with CVP, and the same immunization process was repeated after one month. After a period of 15 months following the initial immunization, mice were infected with 1 × 10^5^ PFU/50 µL/animal of SARS-CoV-2 early circulating strain (TY/WK-521) or Omicron BA.5 variant (TY41-702).

### 2.8. Quantification of Viral RNA

Mouse lungs and trachea were harvested and homogenized in nine volumes of Leibovitz’s L-15 medium (Gibco, Waltham, MA, USA #11415-064) using a multi-bead shocker (Yasui Kikai, Osaka, Japan). Total RNA was then extracted from the homogenates using an RNeasy Mini Kit (Qiagen, #74106) according to the manufacturer’s instructions. The quantification of viral RNA was performed by qRTP–CR. The following primers and probe were used to detect the WK-521 strain: forward primer, 5′-GACCCCAAAATCAGCGAAAT-3′; reverse primer, 5′-TCTGGTTACTGCCAGTTGAATCTG-3′; and probe, 5′-(FAM)-ACCCCGCATTACGTTTGGTGGACC-(BHQ-1)-3′. The following primers and probe were used to detect the TY38-873 and TY41-702 strains: forward primer, 5′-TTACAAACATTGGCCGCAAA-3′; reverse primer, 5′-GCGCGACATTCCGAAGAA-3′; and probe, 5′-(FAM)-ACAATTTGCCCCCAGCGCTTCAG-(BHQ-1)-3′. FAM and BHQ-1 denote 6-fluorescein amidite and Black Hole Quencher-1, respectively.

### 2.9. Determination of Viral Titer

The supernatant acquired from the mouse lung homogenate was subjected to serial 10-fold dilutions. The diluted samples were added to TMPRSS2-expressing Vero cells in a 6-well plate and incubated for 1 h. The samples were then removed from the plate and overlaid using a growth medium containing 0.6% agarose and incubated for 2–3 days. The plate was fixed using formalin and the plaques were visualized following crystal violet staining.

### 2.10. Histopathological and Immunohistochemical Analyses of Lungs

Mouse lungs were fixed in formalin and embedded in paraffin, and sections were prepared at a thickness of 4 µm. For immunohistochemical evaluation, deparaffinized sections immersed in 10 mM citrate buffer (pH 6.0) were antigen retrieved by autoclaving at 121 °C for 10 min. To inactivate the endogenous peroxidases, the sections were incubated in 1% hydrogen peroxide in methanol for 20 min at RT. Sections were then blocked with BlockAce (DS Pharma Biomedical, Osaka, Japan) for 20 min at RT. Subsequently, the tissue samples were incubated with SARS-CoV-2 N protein–specific antibody ([HL344], 1:1000, GeneTex, Irvine, CA, USA) diluted in 0.25% Tween-20 PBS (−) overnight at 4 °C. Samples were then incubated with the EnVision+ System HRP-labeled Polymer Anti-Rabbit (Dako, Santa Clara, CA, USA) for 1 h at RT. Antigens detection was performed using DAB chromogen with ImmPACT DAB Peroxidase Substrate (Vector Laboratories, Newark, CA, USA). Cell nuclei were stained with a hematoxylin solution.

### 2.11. Depletion of Immunocompetent Cells

Alveolar macrophages were depleted using clophosome-N (FormuMax Scientific, Sunnyvale, CA, USA #F70 clophosome-N101C-N) intranasally administered to mice 2 days before the infection with SARS-CoV-2. Ly6C^hi^ monocytes and Ly6G^+^ neutrophils were depleted using 250 µg of anti-Gr-1 mAb (RB6-8C5, rat IgG2b) intraperitoneally injected 2 days before infection and the injection was repeated every third day. CD4^+^ or CD8^+^ cells were depleted using 50 μg of anti-CD4 (GK1.5, Bio X Cell, Lebanon, NH, USA #BE0003-1) or anti-CD8 (2.43, Bio Cell #BE0061) mAb, intravenously injected on the day of infection, followed by repeat administration on every third day.

### 2.12. Statistical Analysis

Statistical analyses were carried out with Prism (version 8.3.0; GraphPad). Results are expressed as mean ± standard deviation (SD) or geomean ± geometric standard deviation. *p* values were calculated using two-tailed, non-paired Student’s *t*-tests for data acquired from two groups; one-way ANOVA or two-way ANOVA followed by post hoc Dunnett’s or Tukey’s multiple comparison test was used, as appropriate, for data from more than two groups/variables. Statistical significance was considered at *p* < 0.05. The correlation coefficient (*r*) was calculated by using the appropriate function in Prism.

## 3. Results

### 3.1. Intranasally Inoculated S–CVP Induces High Levels of Antigen-Specific Antibodies and Neutralizing Antibodies

The effects of intranasal inoculation of S protein of SARS-CoV-2 mixed with CVP (S–CVP) on antibody induction were evaluated. BALB/c mice immunized with S–CVP exhibited high antigen-specific IgG and IgA levels in the serum (Figure 1A); moreover, IgA was detected in both BALF and nasal lavage fluid (Figure 1B). Inoculation with S-alum induced serum IgG only, however, this level was significantly lower than that induced by S–CVP. Furthermore, inoculation with S–CVP elicited neutralizing antibodies in serum and BALF (Figure 1C). Some mice that received S-alum exhibited neutralizing antibodies only in the serum, which was not detected in the mucosal epithelium. In contrast, no binding and neutralizing antibodies were detected in mice that received S or the vehicle alone. These findings indicate that intranasal inoculation with S–CVP enhanced the production of binding and neutralizing antibodies in the mucosal epithelium and serum of mice.

### 3.2. Intranasally Inoculated S–CVP Induces a Strong Cellular Immune Response

Next, we examined the effectiveness of S–CVP in the induction of cellular immune responses. The ELISpot assay revealed significantly enhanced secretion of IFN-γ and IL-2, but no significant increase in IL-4 was observed in mice inoculated with S–CVP compared with than in vehicle-inoculated mice (Figure 1D). However, mice vaccinated with S-alum produced high levels of IL-4. In contrast, cytotoxic T-cell response was not detected in mice inoculated with S alone or with the vehicle. These data suggest that intranasal inoculation with S–CVP induces a robust Th1-prone T-cell response.

### 3.3. S–CVP Induces Protective Immunity against SARS-CoV-2 Challenge

We assessed the protective effects of S–CVP against SARS-CoV-2 infection in mice. Mice were inoculated intranasally with S protein, either with or without CVP (S–CVP or S alone). As a control, a mixture of alum and S was administered subcutaneously in either a single or double dose. At 6 weeks after inoculation, mice were infected with the SARS-CoV-2 early pandemic strain (TY/WK-521). Five days post-infection, the mice were euthanized, and samples were collected for further analysis (Figure 2A). Mice inoculated with S–CVP or a double dose of S-alum exhibited a transient reduction in body weight. In contrast, excessive body weight loss was detected in the other groups (Figure 2B). Viral RNA loads were significantly lower in mice vaccinated with S–CVP than in those treated with the vehicle or S alone and were approximately ten times lower than that recorded in the double dose S-alum (Figure 2C). Moreover, the infectious SARS-CoV-2 titer in the lungs of S–CVP-inoculated mice was below the detection limit except in one (Figure 2D). However, infectious particles were detected in 100% and 60% of the mice in the S-alum single and double-dose groups, respectively. Furthermore, mice vaccinated with S–CVP showed significantly higher levels of IFN-γ production compared to those vaccinated with the vehicle, whereas no significant increase in IL-4 production was observed (Figure 2E). Immuno-histological staining of the SARS-CoV-2 N protein revealed a minimal SARS-CoV-2 antigen in the S–CVP vaccinated groups, whereas a high abundance of viral antigen was detected in the alveolar epithelial cells of the other groups (Figure 2F). These data indicate that inoculated S–CVP reduces the SARS-CoV-2 viral load in the upper and lower respiratory tracts of mice.

### 3.4. S–CVP Suppresses Lung Inflammation during SARS-CoV-2 Infection

Next, we examined the effects of S–CVP on lung inflammation during SARS-CoV-2 infection. Histopathological analyses revealed that mice vaccinated with the vehicle or a single dose of S-alum exhibited moderate pneumonia characterized by inflammatory cell infiltration (Figure 3A). In contrast, thickened alveolar walls with suppressed inflammatory cell infiltration were observed in mice that received S–CVP, S alone, or a double dose of S-alum.

We analyzed the effect of S–CVP on cytokine production. In the vehicle-treated mice, SARS-CoV-2 infection enhanced cytokines and chemokines production, including IL-6, tumor necrosis factor (TNF)-α, granulocyte colony-stimulating factor (G-CSF), IFN-γ, and monocyte chemotactic protein (MCP)-1, which play important roles in the development of hyper-inflammation (Figure 3B) [15]. Contrastingly, considerably lower levels of these cytokines were identified in mice inoculated with S–CVP or S-alum. The S-alum-inoculated group showed significantly higher levels of IL-4 and IL-5, typical Th2 cytokines, whereas the levels of these cytokines were not increased in the S–CVP-inoculated group (Figure 3C). These results indicated that inoculation with S–CVP suppresses lung inflammation.

### 3.5. Intranasal Inoculation with S–CVP Provides Cross-Protection against the Omicron BA.1 Variant in Mice

The cross-protective effect of S–CVP against the Omicron BA.1 variant was examined using the previously described method. Mice inoculated with S–CVP or S-alum exhibited reduced amounts of viral RNA and infectious particles in the lungs compared to the vehicle or S alone (Figure 4A). Furthermore, infectious viruses were detected in all mice inoculated with S-alum; however, they were reduced below the detection limit in all S–CVP-treated mice except one (Figure 4B). These results indicate that S–CVP has cross-protection against the Omicron BA.1 variant.

We examined the immune responses elicited by S–CVP against factors associated with cross-reactivity. PRNT_50_ titers were barely detected in mice inoculated with S–CVP (Figure 4C) and were not associated with virus clearance. In contrast, the serum IgG and IgA titers in mice vaccinated with S–CVP were comparable between Omicron BA.1 and early pandemic strain-derived S protein (Figure 4D). However, inoculation with S-alum significantly reduced serum IgG titers against Omicron BA.1-derived S compared to those against the early pandemic strain, and IgA was not detectable. Furthermore, S–CVP-inoculated mice significantly produced IFN-γ against the homologous SARS-CoV-2 as well as the Omicron BA.1 variant (Figure 4E). Cellular immunity and binding antibodies against SARS-CoV-2 Omicron BA.1 variant were detected in S–CVP-inoculated mice. Therefore, to validate the contribution of these factors to the clearance virus, we depleted the cells associated with these immune responses during infection. In S–CVP-inoculated mice with depleted Ly6G/Ly6C (Gr-1) positive (neutrophils and monocytes) or CD8α^+^ (CD8^+^ T-cells) cells, the ability to clear the viruses was inhibited compared to the mice with no depletion (Figure 4F). Clophosome-N administration, which depletes macrophages, exhibited no comparable effect. Furthermore, viral elimination was more severely impaired in the lungs of mice in which CD4^+^ and CD8α^+^ cells were simultaneously depleted. While exploring the association between the depletion of these factors and antibody production after SARS-CoV-2 infection, we found that none of these factors were required for antibody production after infection (Figure 4G). These results suggest that S–CVP inoculation protects mice against the SARS-CoV-2 Omicron BA.1 variant; moreover, binding antibodies and cellular immunity play an important role in viral clearance.

### 3.6. Intranasal Inoculation with S–CVP Induces Long-Lasting Immunity in Mice

The induction of long-lasting immunity is a key objective in vaccine development. Therefore, we analyzed the persistence of the immunity induced by S–CVP (Figure 5A). The titers of homologous S antigen-specific IgG and IgA showed a greater increase after the second immunization and were maintained at a high level without attenuation even after 15 months (Figure 5B); a strong cross-reactivity with the Omicron BA4/5 variant-derived S was identified. After a period of 15 months following vaccination, RPNT_50_ titers against the early circulating strain were as high as those recorded after 1 month of additional vaccination (Figure 5C). However, PRNT_50_ titers against the Omicron BA.5 variant were almost undetectable.

Next, we examined whether this long-term immunity effectively protected mice against SARS-CoV-2 (Figure 5A). The S–CVP-vaccinated mice showed a significant reduction in infectious SARS-CoV-2 load (both the early circulating strain and Omicron BA.5 variant) (Figure 5D). These findings were further validated through immunostaining assay for viral antigens using lungs (Figure 5E, upper panel). Histopathological analysis of the untreated and S-inoculated groups showed pneumonia, associated cell infiltration, and thickening of the alveolar walls, whereas inflammation was suppressed in the S–CVP-inoculated group (Figure 5E, lower panel). Therefore, immunity elicited by S–CVP persisted for at least 15 months and provided effective immunity against mutant strains of SARS-CoV-2.

## 4. Discussion

This study demonstrated that intranasally inoculated S–CVP can induce strong humoral (Figure 1A,B) and cellular (Figure 1D) immune responses and protect mice from SARS-CoV-2 infection. Furthermore, S–CVP-induced long-lasting immunity in mice effectively suppressed the load of the SARS-CoV-2 Omicron BA.5 variant even 15 months after immunization (Figure 5D). The comparative analyses confirmed that CVP enhances the immune response induced by intranasal vaccination.

Intranasal vaccines are considered advantageous because they can induce both mucosal and systemic immunity while being minimally invasive. Mucosal immunity is the first line of defense against respiratory tract infections, and IgA secreted by the mucosa plays an important role in this immunity [16,17]. However, intranasal immunization requires a strong adjuvant, partly because clearance of the antigen by host defense mechanisms is relatively rapid, whereas the antigenicity is low. Double-stranded RNA and CpG oligodeoxynucleotides, ligands for Toll-like receptors, have been reported to be potent mucosal vaccine adjuvants [18,19]. Currently, CVP is used as a thickening agent in pharmaceutical production and, hence, its safety is well established. Owing to its high viscosity, CVP is considered to prolong the residence time of antigens in the nasal cavity, facilitating efficient uptake by antigen-presenting cells, and inducing a robust immune response; however, the precise mechanism behind this is unknown.

SARS-CoV-2 is a single-stranded RNA virus that mutates rapidly. The Omicron variant, with more than 30 substitutions in the amino acid sequence of the S protein, can escape several existing neutralizing antibodies, leading to the reduced protective effect of vaccines [20]. However, mice inoculated with S–CVP also showed rapid clearance of the Omicron BA.1 variant (Figure 4A,B). Therefore, we investigated the factors that contribute to the S–CVP-induced clearance of SARS-CoV-2 mutant virus; viral clearance was inhibited by the use of anti-Gr1 [21] or anti-CD8 antibodies (Figure 4F). Anti-Gr-1 antibodies reportedly bind to both Ly6C and Ly6G [22], suggesting that the administration of anti-Gr-1 antibodies depletes neutrophils and monocytes. Neutrophils and monocytes express FcRIIIa or FcRIIIb and participate in viral clearance via antibody-dependent cell-mediated phagocytosis [23,24]. However, these results are insufficient to determine the effectiveness of neutrophils and/or monocytes in viral clearance. A previous study reported that administration of an anti-Gr-1 antibody suppressed SARS virus clearance, whereas specific depletion of neutrophils did not [25]. Therefore, antibody-dependent virus clearance by phagocytic cells such as monocytes is effective against the Omicron BA.1 variant. CD8^+^ cells play a chief role in cellular immunity and critically influence the clearance of viruses [26]. Recently, memory T-cells in the mucosa were demonstrated to provide long-term protection [27]. The cellular immunity induced by S–CVP was cross-reactive with the Omicron BA.1 variant (Figure 4E). Moreover, the depletion of CD8^+^ cells alone inhibited viral clearance; however, the simultaneous depletion of CD4^+^ and CD8^+^ cells caused a further increase in the viral load (Figure 4F), indicating the significant role of CD4^+^ T-cells in viral clearance. The post-infection antibody titer in the CD4-depleted group was comparable to that in the untreated group, which suggests a role of CD4^+^ T-cells, in addition to the enhancement of antibody production. Although CD4^+^ T-cells enhance the activity of CD8^+^ T-cells, several studies have indicated that CD4^+^ T-cells exhibit cytotoxic activity [28]. Transferring activated CD4^+^ T-cells protects mice from lethal influenza infections [29]. CD4^+^ T-cells induced by S–CVP are considered to exhibit cytotoxic properties, which act together with CD8^+^ T-cells and contribute to virus elimination (Figure 4F). Thus, our results suggest that the potent virus-reducing effect of S–CVP is potentially attributable to its ability to induce cell-mediated immunity, in addition to humoral immunity.

mRNA-based vaccines used worldwide are highly efficient against COVID-19 by strongly inducing neutralizing antibodies and cellular immunity [4,30]. However, the protective effects gradually decline after vaccination [31]. Hence, additional doses at intervals of several months are recommended. Therefore, the development of vaccines with long-lasting effects is urgently needed. Intranasal immunization using S–CVP maintained the levels of binding and neutralizing antibodies for more than 15 months. Furthermore, this strategy showed strong cross-reactivity for the SARS-CoV-2 Omicron BA.5 variant and alleviated the infection, even when the S-antigen derived from the early circulating strain was used for immunization. Thus, intranasal inoculation with CVP and antigens may provide a vaccine with long-lasting and broad-spectrum effects.

In this study, we demonstrated that S–CVP may be a promising candidate for an intranasal vaccine in a mouse model. Hereafter, further challenging studies in non-human primates and humans are required to evaluate the practical potential of this vaccine. Several intranasal vaccines against SARS-CoV-2, the efficacy of which have been demonstrated in animal models, are currently undergoing clinical trials [32]. However, not all of them will yield the expected results in humans. For example, intranasal vaccines using viral vectors have not shown sufficient immune responses in phase I trials [33]. It is speculated that the expression levels of host receptors on mucosal epithelium and professional antigen-presenting cells, as well as the infectivity of the vector, influenced the outcomes. CVP is a viscous agent and may not be affected by host range, such as viral vector. Furthermore, this study used only the combined antigen and CVP for intranasal immunization and did not compare other additives used in intranasal vaccines. Future studies are needed to investigate whether the long-lasting efficacy of S–CVP against a wide range of virus variants observed with intranasal immunization is superior to that of other intranasal vaccines.

In summary, intranasal inoculation using the recombinant S protein of SARS-CoV-2 combined with CVP strongly induces antibody production in the serum and mucosa of mice and enhances cellular immunity. Furthermore, this strategy effectively induced a long-lasting immunity against a wide range of SARS-CoV-2 variants, ranging from early pandemic strains to Omicron BA.5. Therefore, we suggest that S–CVP as a useful vaccine. In this established method, recombinant proteins have been used as antigens; hence, this strategy can be simply modified by changing the antigens to immunize the hosts against different viruses.

## Figures and Tables

**Figure 1 vaccines-12-00794-f001:**
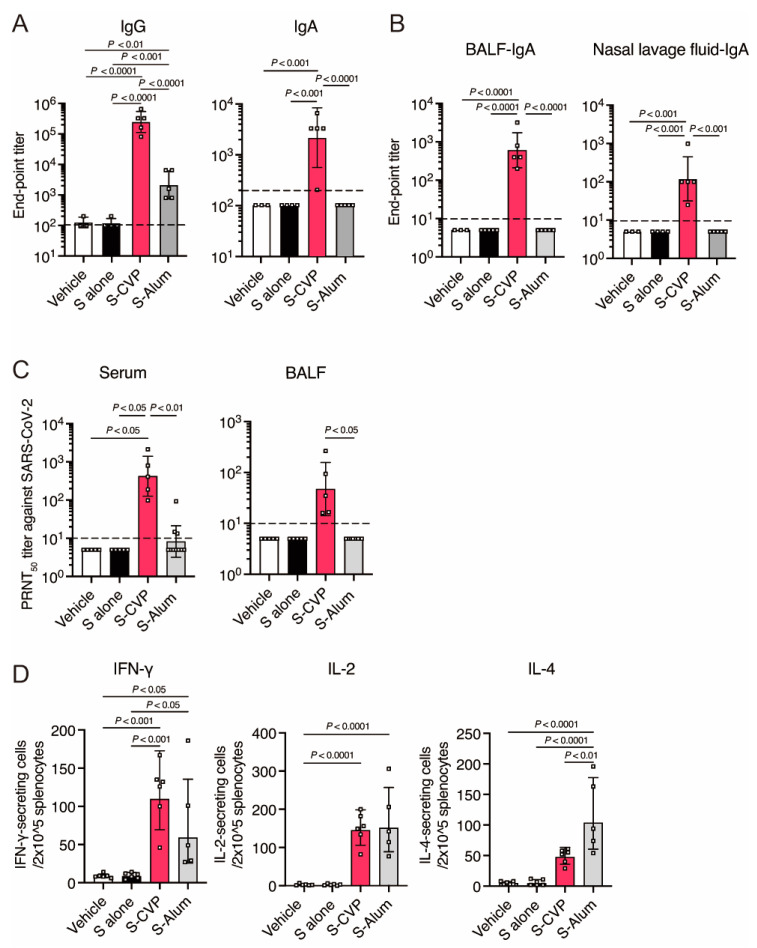
Humoral and cell-mediated immune responses induced by intranasally inoculated S–CVP. Three weeks after vaccination, mice were euthanized and samples were collected. (**A**) SARS-CoV-2 S-specific IgG (**left**) and IgA (**right**) in mouse serum were detected by ELISA. ‘Vehicle’ means administration of CVP only. (**B**) SARS-CoV-2 S-specific IgA in mouse BALF (**left**) and nasal lavage fluid (**right**) were detected by ELISA. (**C**) Neutralizing antibody titer of mouse serum (**left**) or BALF (**right**) against SARS-CoV-2 early pandemic strain (TY/WK-521) was measured. (**D**) IFN-γ (**left**), IL-2 (**middle**), and IL-4 (**right**) ELISpot assays using splenocytes. In all panel, each dot represents data from an individual mouse. (**A**–**C**) Values are presented as the geometric mean with geometric SD. Dashed lines indicate the limit of detection (LOD). Values below the LOD are shown as a half value of LOD. (**D**) Values are shown as the mean ± SD. Statistical analysis was performed using two-tailed one-way ANOVA with post hoc Tukey’s multiple comparison tests.

**Figure 2 vaccines-12-00794-f002:**
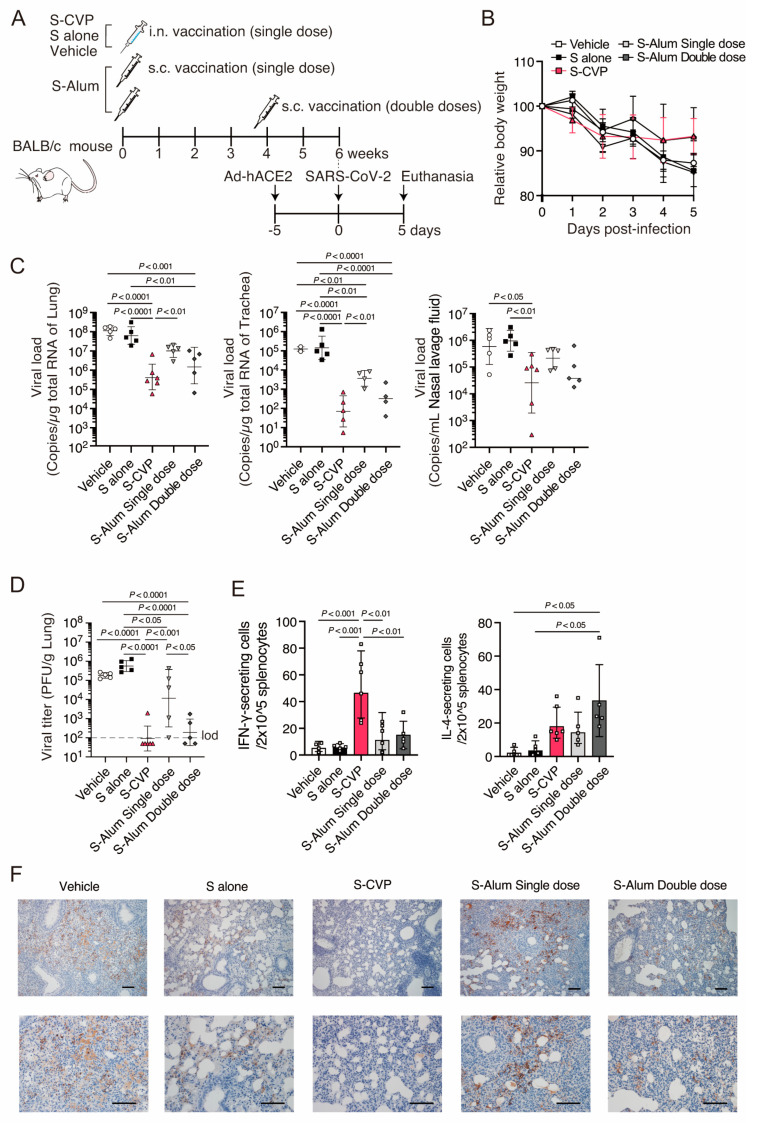
Protective efficacy of intranasally inoculated S–CVP against homologous SARS-CoV-2 infection in mice. (**A**) Schematic illustration presenting the study design. (**B**) Body weight changes were monitored following the SARS-CoV-2 challenge. (**C**) Viral load in mouse lung (**left**), trachea (**middle**), and nasal lavage fluid (**right**) was measured using qRT–PCR. (**D**) Viral titer in mouse lungs was measured through plaque assay. The dashed line indicates the LOD. Viral titers below the LOD are shown as half value of LOD. (**E**) IFN-γ (**left**) or IL-4 (**right**) ELISpot assay using splenocyte. (**F**) Immunohistochemical analysis of SARS-CoV-2 nucleocapsid (N) antigen in lung sections; the original magnification was 100× (upper row) or 200× (lower row). (**C**–**E**) Each dot represents data from an individual mouse. (**C**,**D**) Black bar indicate the geometric mean with geometric SD. (**B**,**E**) Values are shown as the mean ± SD. Statistical analysis was performed by two-tailed one-way ANOVA with post hoc Tukey’s multiple comparisons tests.

**Figure 3 vaccines-12-00794-f003:**
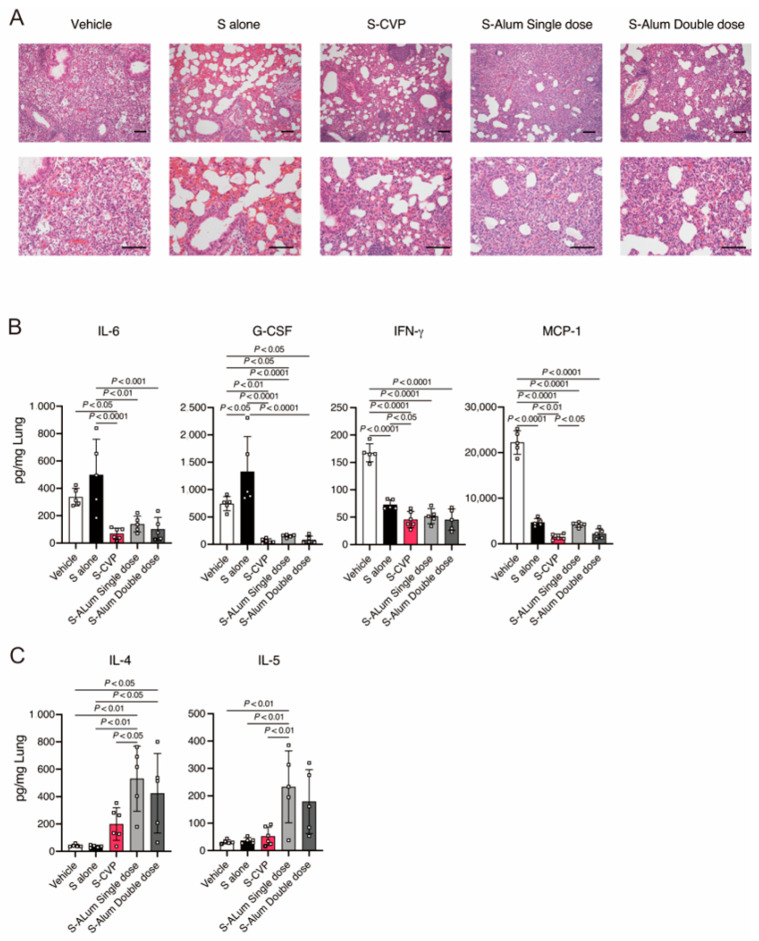
Intranasally inoculated S–CVP suppressed lung inflammation during SARS-CoV-2 infection. (**A**) Mouse lung tissues were analyzed using hematoxylin and eosin staining. The original magnification was 100× (upper row) or 200× (lower row). (**B**,**C**) Concentrations of cytokines and chemokines in mouse lungs were measured using a multi-plex bead array. Each dot represents data from an individual mouse and values are presented as mean ± SD. Statistical analysis was performed using two-tailed one-way ANOVA with post hoc Tukey’s multiple comparison tests.

**Figure 4 vaccines-12-00794-f004:**
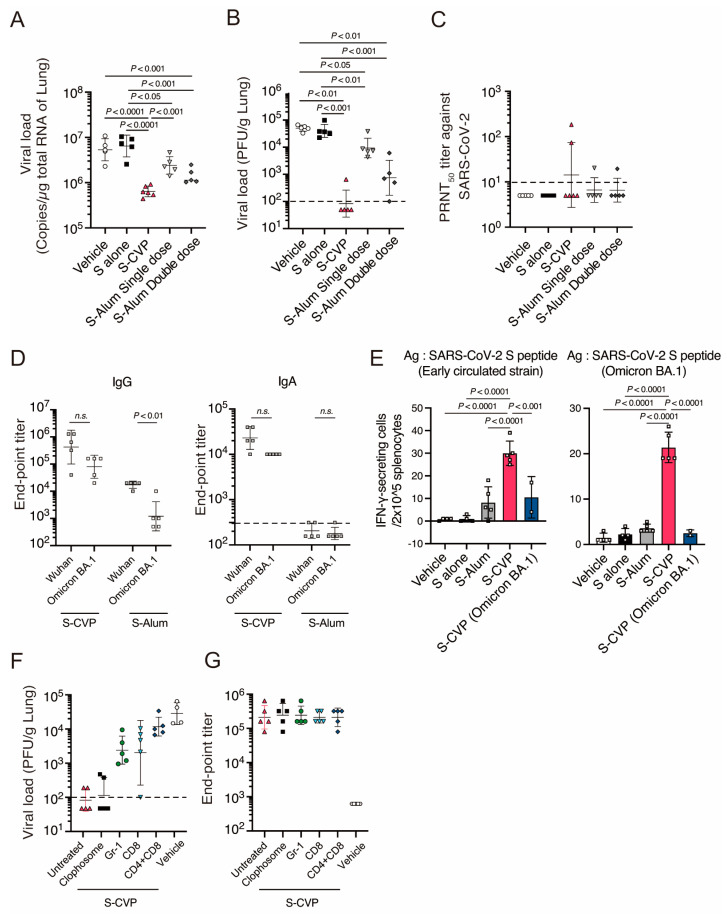
Intranasally inoculated S–CVP reduced virus titer in mice infected with SARS-CoV-2 Omicron BA.1 variant. The viral load in mouse lungs was measured using (**A**) qRT–PCR and (**B**) plaque assays. (**C**) Titers of neutralizing antibodies against the SARS-CoV-2 Omicron BA.1 variant were measured in mouse serum using PRNT_50_. (**D**) Serum SARS-CoV-2 S derived from the early pandemic strain or Omicron BA.1-specific IgG (**left**) or IgA (**right**) was measured using ELISA. (**E**) IFN-γ ELISpot following stimulation of splenocyte samples with PepTivator SARS-CoV-2 Prot-S complete (**left**) and PepTivator SARS-CoV-2 Prot-S B.1.1.529 WT (**right**). (**F**) Viral load in mouse lungs depleted of immune cells was measured using a plaque assay. (**G**) SARS-CoV-2 Omicron BA.1-derived S-specific IgG in mouse serum was measured using ELISA. In all panel, each dot represents data from an individual mouse. (**A**–**D**,**F**,**G**) Black bar indicate the geometric mean with geometric SD. The dashed lines indicate the LOD. Values below the LOD are depicted as half of the LOD. (**E**) Values are expressed as the mean ± SD. (**A**–**C**,**E**) Statistical analysis was performed using two-tailed one-way ANOVA with post-hoc Tukey’s multiple comparison tests. n.s. = not significant.

**Figure 5 vaccines-12-00794-f005:**
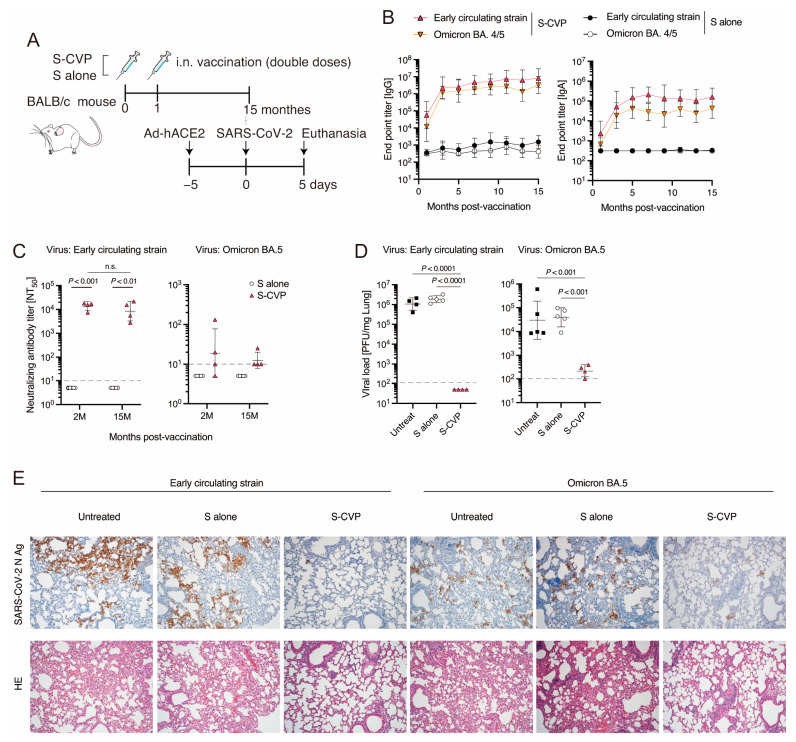
Intranasally inoculated S–CVP induced long-lasting immunity. (**A**) Schematic illustration of the study design. (**B**) SARS-CoV-2 S-specific IgG (**left**) and IgA (**right**) in mouse serum were measured using ELISA. (**C**) PRNT_50_ titers against SARS-CoV-2 early pandemic strain (**left**) or Omicron BA.5 variant (**right**) in mouse serum were measured. (**D**) Viral titer in mouse lungs was measured using plaque assay. (**E**) Immunohistochemical analysis for SARS-CoV-2 nucleocapsid (N) antigen (upper row) or H and E staining (lower row) assay of lung tissues. The original magnification was 100×. (**B**) Values are expressed as the geometric mean with geometric SD. (**C**,**D**) Each dot represents data from an individual mouse and black bar indicate the geometric mean with geometric SD. Dashed lines indicate the LOD. Values below the LOD are depicted as half of the LOD+. Statistical analysis was performed using two-tailed one-way ANOVA with post hoc Tukey’s multiple comparison tests. n.s. = not significant.

## Data Availability

Data are contained within the article.

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
