# Peer review of "Intranasally Inoculated SARS-CoV-2 Spike Protein Combined with Mucoadhesive Polymer Induces Broad and Long-Lasting Immunity"

_vaccines, 2024, doi:10.3390/vaccines12070794_

Round 1

Reviewer 1 Report

Comments and Suggestions for Authors

The study entitled “Intranasally inoculated SARS-CoV-2 spike protein combined with mucoadhesive polymer induces broad and long-lasting immunity” is generally well-written and shows robust data regarding the protection elicited by an intranasal vaccine composed of SARS-CoV-2 trimeric S protein + CVP in mice. Although this study is reasonable and promising, there are some points that I would like to ask the authors to improve.

Major points:

1-     To explain why you did not perform intranasal administration of (Alum+S protein) and/or subcutaneous administration of (CVP+S protein) as controls.

2-     Please make sure to improve the schematic illustrations of the vaccine regimen to clearly show the dosage, the total number of doses, the timing of administration, and whether it was administered subcutaneously or intranasally. It also needs to be done throughout the results and/or subtitles.

3-     To clarify what is the vehicle. If it is CVP, I suggest changing the information on figures or explaining within the legends;

4-     In my opinion, the conclusion in lines 234 and 235 is somehow an extrapolation, as CVP and Alum were not compared when administered through the same route. I would like authors to remove it or perform experiments using S-CVP administered for s.c. pathway to make a reasonable comparison;

5-     The figures need to be improved in quality, at least 300 d.p.i. Some of them are bad in quality such as the immunohistochemistry panels.

6-     To quantify the alveolar space to compare better the groups regarding lung inflammation.

7-     I expected that authors have discussed the use of intranasal vaccines in humans. We already have an intranasal vaccine approved against Influenza, but clinical trials for COVID-19 intranasal vaccines did not show promising results. For instance, the clinical trial conducted with the Chadox1 vaccine. In this regard, what is the chance of this vaccine being effective for humans? Studies have demonstrated protection in mice but failed in humans. It can be included as limitations and future perspectives in the final of the discussion topic. Furthermore, I wanted the authors to discuss the overall limitations of the study, maybe one is the absence of comparison with other adjuvants used intranasally.

Minor points:

1-     To include a reference in the first paragraph, lines 38 to 45;

2-     What does SLC mean? To write in full. (Line 76);

3-     I suggest the authors include the pandemic strain and variant names in the 2.4 topic instead of only in 2.5;

4-     To write LOD in full (Line 214);

5-     I suggest inserting figure 1 after topic 3.2 or putting topics 3.1 and 3.2 together;

6-     Remove the first “are” in line 400;

7-     The authors need to include the reference regarding line 395 until line 397.

Comments on the Quality of English Language

The study is well-written and the English is satisfactory. 

Author Response

Reviewer #1

The study entitled “Intranasally inoculated SARS-CoV-2 spike protein combined with mucoadhesive polymer induces broad and long-lasting immunity” is generally well-written and shows robust data regarding the protection elicited by an intranasal vaccine composed of SARS-CoV-2 trimeric S protein + CVP in mice. Although this study is reasonable and promising, there are some points that I would like to ask the authors to improve.

Major points:

  1. To explain why you did not perform intranasal administration of (Alum+S protein) and/or subcutaneous administration of (CVP+S protein) as controls.

The purpose of this study was to determine the efficacy of intranasal inoculation of CVP+S. We employed Alum, a typical adjuvant for protein vaccines, and its common method of inoculation, subcutaneous inoculation. Therefore, no intranasal inoculation of Alum+S or subcutaneous inoculation of CVP+S was performed in this study.

  1. Please make sure to improve the schematic illustrations of the vaccine regimen to clearly show the dosage, the total number of doses, the timing of administration, and whether it was administered subcutaneously or intranasally. It also needs to be done throughout the results and/or subtitles.

Thank you for your suggestion. According to this reviewer’s comment, we have improved the schematic illustrations (Figure 2A, Figure 5A). In addition, a description of the method of vaccine administration and number of doses was included in the Results section of the revised manuscript. (3.3; p6, line 202-206).

[Mice were inoculated intranasally with S protein, either with or without CVP (S-CVP or S alone). As a control, a mixture of alum and S was administered subcutaneously in either a single or double dose. At 6 weeks after inoculation, mice were infected with the SARS-CoV-2 early pandemic strain (TY/WK-521). Five days post-infection, the mice were euthanized, and samples were collected for further analysis (Figure 2A). ]

  1. To clarify what is the vehicle. If it is CVP, I suggest changing the information on figures or explaining within the legends;

Based on this reviewer’s comment, we have added to the legends that vehicle refers to CVP inoculation (p6, line 193).

[‘Vehicle’ means administration of CVP only.]

  1. In my opinion, the conclusion in lines 234 and 235 is somehow an extrapolation, as CVP and Alum were not compared when administered through the same route. I would like authors to remove it or perform experiments using S-CVP administered for s.c. pathway to make a reasonable comparison;

In keeping with this reviewer’s comment, we deleted this sentence from the revised manuscript (Therefore, S-CVP is considered more effective than S-Alum at eliminating viruses).

  1. The figures need to be improved in quality, at least 300 d.p.i. Some of them are bad in quality such as the immunohistochemistry panels.

Due to the file size being too large to upload through the submission portal, the resolution of the image has been temporarily reduced. We have already prepared a 300 d.p.i. image and will replace it.

  1. To quantify the alveolar space to compare better the groups regarding lung inflammation.

As suggested by the reviewer, we compared the area of alveolar space in each group. This was done by taking 5 random images from HE-stained images of lung sections, calculating the area of alveolar space for each image, and showing the mean value as a single dot. The bar graph shows the average value ± SD for each group. The results show that the S-CVP vaccinated group may appear to have a larger alveolar space area than the other groups.

  1. I expected that authors have discussed the use of intranasal vaccines in humans. We already have an intranasal vaccine approved against Influenza, but clinical trials for COVID-19 intranasal vaccines did not show promising results. For instance, the clinical trial conducted with the Chadox1 vaccine. In this regard, what is the chance of this vaccine being effective for humans? Studies have demonstrated protection in mice but failed in humans. It can be included as limitations and future perspectives in the final of the discussion topic. Furthermore, I wanted the authors to discuss the overall limitations of the study, maybe one is the absence of comparison with other adjuvants used intranasally.

Based on this reviewer’s comment, we have added a passage to the Discussion section of the revised manuscript regarding the considered efficacy for humans and the limitations of this study (p16, line 360-370).

[In this study, we demonstrated that S-CVP may be a promising candidate for an intranasal vaccine in a mouse model. Hereafter, further challenging studies in non-human primates and humans are required to evaluate the practical potential of this vaccine. Several intranasal vaccines against SARS-CoV-2, the efficacy of which has been demonstrated in animal models, are currently undergoing clinical trials 31. However, not all of them will yield the expected results in humans. For example, intranasal vaccines using viral vectors have not shown sufficient immune responses in phase I trials 32. It is speculated that the expression levels of host receptors on mucosal epithelium and professional antigen-presenting cells, as well as the infectivity of the vector, influenced the outcomes. CVP is a viscous agent and may not be affected by host range, such as viral vector. Furthermore, this study used only the combined antigen and CVP for intranasal immunization and did not compare other additives used in intranasal vaccines. Future studies are needed to investigate whether the long-lasting efficacy of S-CVP against a wide range of virus variants observed with intranasal immunization is superior to that of other intranasal vaccines.]

Minor points:

  1. To include a reference in the first paragraph, lines 38 to 45;

Based on this reviewer’s comment, we inserted supporting references for this passage (p1, line 39, Reference 6: Levin, E. G. et al. New England Journal of Medicine 385, e84 (2021)).

  1. What does SLC mean? To write in full. (Line 76);

This is the name of the vendor from which the mice were procured. Based on the reviewer’s comment, we have changed the notation to ‘Japan SLC, Inc.’ (p2, line 68).

  1. I suggest the authors include the pandemic strain and variant names in the 2.4 topic instead of only in 2.5;

In response to a reviewer’s suggestion, we have shifted the description of the pandemic strain and variant names to section ‘2.4’ from ‘2.5’ (p3 line 88-90).

[SARS-CoV-2 JP/TY-WK-521/2020 (WK-521; Accession: LC522975, early pandemic strain), SARS-COV-2 hCoV-19/Japan/TY38-873/2021 (TY38-873; GISAID strain name: EPI_ISL_7418017, Omicron BA.1 variant), and SARS-CoV-2 hCoV-19/Japan/TY41-702 (Omicron BA.5 variant), kindly provided by Drs. Masayuki Saijo, Mutsuyo Takayama-Ito, Masaaki Sato, and Ken Maeda, National Institute of Infectious Disease, were used as challenge strains.]

  1. To write LOD in full (Line 214);

Based on this reviewer’s comment, we included the definition of the abbreviation ‘LOD’ (Limit of Detection) in the legend (p6, line 197).

  1. I suggest inserting figure 1 after topic 3.2 or putting topics 3.1 and 3.2 together;

In response to a reviewer’s suggestion, we have changed the position of Figure 1 to follow subsection 3.2 of the manuscript.

  1. Remove the first “are” in line 400;

Based on this reviewer’s comment, we modified the sentence in the revised manuscript (p15, line 352-353).

[mRNA-based vaccines used worldwide are highly efficient against COVID-19 by strongly inducing neutralizing antibodies and cellular immunity]

  1. The authors need to include the reference regarding line 395 until line 397.

Based on this reviewer’s comment, we inserted the corresponding figure number (p15, line 349).

Reviewer 2 Report

Comments and Suggestions for Authors

The authors immunized female mice intranasally with SARS-CoV-2 spike (S) protein mixed with a carboxyvinyl polymer (CVP) once (or twice for some experiments). Compared to controls (S alone and S with Alum), S-CVP induced both IgG and IgA responses, and the controls did not induce IgA response. The induced antibodies neutralized the wild-type virus but not the Omicron variants; however, the immunized mice were protected from Omicron challenges. Further analysis indicated that S-CVP induced cellular responses and possibly antibody Fc-effector functions, both required for mice protection. Additionally, S-CVP x2 induced immunity and mice protection that lasted for 15 months after the initial immunization. The authors concluded that intranasal immunization with S-CVP is a promising vaccine strategy against SARS-CoV-2. Overall, the data supported the conclusion. Below are points requiring further clarifications.

1.    Page 1 Line 34, vaccines did not “control” SARS-CoV-2 infection. Vaccines helped “mitigate” the impact of SARS-CoV-2 infection.

2.    Page 1 Line 35, should be "prophylactic efficacy in terms of severe disease development."

3.    Page 2 Line 51, many antigens “are” rapidly removed.

4.    Page 2 Line 78, the amount of CVP was not described.

5.    Page 3 Line 116, were ELISpot samples from mice immunized twice and other assays from mice immunized once? Why were there differences in immunization times: once vs twice?

6.    The sample time points were not described for Figure 1.

7.    Page 6 Line 222, “T-cell response was not detected”. Should it be “cytotoxic” T-cell response was not detected?

8.    Page 7, Figure 2B, the group symbols did not match; Figure 2E legend said IL-2 but 2E showed IL-4.

9.    Page 8 Line 259, is “thickened alveolar walls” a concerning sign for pathogenesis? Similarly, Page 12 Line 337, also mentioned “thickening of the alveolar walls”.

10.  Page 10 Lines 298-302, the targeted factor or cell population for each depletion needs to be described.

11.  Page 11, Figure 4F and 4G, what is the last column 0B?

12. At the end of Discussion, it may be worth mentioning that the study is limited to mice only.

Comments on the Quality of English Language

The English needs proofreading.  For example, Page 6, Lines 218-224. 

Author Response

Reviewer #2

The authors immunized female mice intranasally with SARS-CoV-2 spike (S) protein mixed with a carboxyvinyl polymer (CVP) once (or twice for some experiments). Compared to controls (S alone and S with Alum), S-CVP induced both IgG and IgA responses, and the controls did not induce IgA response. The induced antibodies neutralized the wild-type virus but not the Omicron variants; however, the immunized mice were protected from Omicron challenges. Further analysis indicated that S-CVP induced cellular responses and possibly antibody Fc-effector functions, both required for mice protection. Additionally, S-CVP x2 induced immunity and mice protection that lasted for 15 months after the initial immunization. The authors concluded that intranasal immunization with S-CVP is a promising vaccine strategy against SARS-CoV-2. Overall, the data supported the conclusion. Below are points requiring further clarifications.

  1. Page 1 Line 34, vaccines did not “control” SARS-CoV-2 infection. Vaccines helped “mitigate” the impact of SARS-CoV-2 infection.

In response to a reviewer’s suggestion, we changed the word “control” to “mitigate” in the revised manuscript (p1, line 34-35).

[Various vaccines have been developed to mitigate the impact of SARS-CoV-2 infection.]

  1. Page 1 Line 35, should be "prophylactic efficacy in terms of severe disease development."

In response to the reviewer’s suggestion, we changed the sentence in the Introduction section (p1, line 35-36).

[Globally, mRNA vaccines, which induce a robust systemic immune response and exhibit prophylactic efficacy in terms of severe disease development, are used.]

  1. Page 2 Line 51, many antigens “are” rapidly removed.

In keeping with the reviewer’s comment, we modified the sentence in the revised manuscript (p2, line 48).

[However, intranasal vaccines require strong and safe adjuvants, which is attributable to the fact that many antigens are rapidly removed via mucociliary clearance.]

  1. Page 2 Line 78, the amount of CVP was not described.

Based on the reviewer’s comment, we have added a description of the liquid volume of the CVP in the Materials and Methods section (p2, line 70).

  1. Page 3 Line 116, were ELISpot samples from mice immunized twice and other assays from mice immunized once? Why were there differences in immunization times: once vs twice?

As the reviewer has noted, we performed the ELISpot assay using mice that had been vaccinated twice. When we evaluate cellular immunity, we do so 7 days after the last immunization, in order to obtain a clear result. This is the case for all other aspects of the study except ELISpot, where a single immunization is sufficient to obtain a satisfactory result. For this reason, different immunization schedules were used.

  1. The sample time points were not described for Figure 1.

To address this reviewer's comment, we added an explanation in the figure legend column stating that samples were used for analysis 3 weeks post-vaccination (p6, line 191-192).

[Three weeks after vaccination, mice were euthanized and samples were collected.]

  1. Page 6 Line 222, “T-cell response was not detected”. Should it be “cytotoxic” T-cell response was not detected?

In response to the reviewer’s suggestion, we modified the sentence in the revised manuscript (p5, line 187).

[In contrast, cytotoxic T-cell response was not detected in mice inoculated with S alone or the vehicle.]

  1. Page 7, Figure 2B, the group symbols did not match; Figure 2E legend said IL-2 but 2E showed IL-4.

We apologize for the error in the legend. We have changed the legend from "IL-2" to "IL-4" to match the figure (p8, line 224).

[(E) IFN-γ (left) or IL-4 (right) ELISpot assay using splenocyte.]

  1. Page 8 Line 259, is “thickened alveolar walls” a concerning sign for pathogenesis? Similarly, Page 12 Line 337, also mentioned “thickening of the alveolar walls”.

Thickening of the alveolar wall is among the histological changes occurring in the lungs of mice after influenza virus infection, as described in previous papers (Ogiwara, et al. The American Journal of Pathology 184.1 (2014): 171-183), and some human patients also show thickening of the alveolar wall when inflammation occurs in the alveoli (Wijsenbeek, et. al. The Lancet 400.10354 (2022): 769-786). Thus, we think that the thickening of the alveoli is a manifestation of inflammation caused by coronavirus infection.

  1. Page 10 Lines 298-302, the targeted factor or cell population for each depletion needs to be described.

In response to the reviewer’s suggestion, we have added target cell information for each depletion reagent used for the experiment (p11. Line 265--267).

[In S-CVP-inoculated mice with depleted Ly6G/Ly6C (Gr-1) positive (neutrophils and monocytes) or CD8α+ (CD8+ T-cells)cells, the ability to clear the viruses was inhibited compared to the mice with no depletion (Figure 4F). Clophosome-N administration, which depletes macrophages, exhibited no comparable effect.]

  1. Page 11, Figure 4F and 4G, what is the last column 0B?

We apologize for the errors in Figure 4F and 4G. We have corrected 0B to the vehicle (CVP), where it should have originally been written.

  1. At the end of Discussion, it may be worth mentioning that the study is limited to mice only.

Based on the reviewer’s comment, we have added a note to the Discussion section that the implications of this study are limited to mice alone (p16 line 360-362).

[In this study, we demonstrated that S-CVP may be a promising candidate for an intranasal vaccine in a mouse model. Hereafter, further challenging studies in non-human primates and humans are required to evaluate the practical potential of this vaccine.]

Reviewer 3 Report

Comments and Suggestions for Authors

The article under review is of significant interest from the point of view of improving vaccine prevention of COVID-19. The technology for using an intranasal vaccine proposed by the authors is undoubtedly original and can significantly increase the effectiveness and safety of the vaccine against SARS-CoV-2, including new genovariants of the virus. From the point of view of the study design and level of evidence of the results, there are no comments. The work fully meets the requirements of evidence-based medicine at this level of research. There are no significant comments regarding the design of the study or the presentation of the results obtained.

Author Response

Reviewer #3

The article under review is of significant interest from the point of view of improving vaccine prevention of COVID-19. The technology for using an intranasal vaccine proposed by the authors is undoubtedly original and can significantly increase the effectiveness and safety of the vaccine against SARS-CoV-2, including new genovariants of the virus. From the point of view of the study design and level of evidence of the results, there are no comments. The work fully meets the requirements of evidence-based medicine at this level of research. There are no significant comments regarding the design of the study or the presentation of the results obtained.

Thank you for your peer review of our manuscript.